# Structural Analysis and Whole Genome Mapping of a New Type of Plant Virus Subviral RNA: Umbravirus-Like Associated RNAs

**DOI:** 10.3390/v13040646

**Published:** 2021-04-09

**Authors:** Jingyuan Liu, Elizabeth Carino, Sayanta Bera, Feng Gao, Jared P. May, Anne E. Simon

**Affiliations:** 1Department of Cell Biology and Molecular Genetics, University of Maryland College Park, College Park, MD 20742, USA; jliu1021@terpmail.umd.edu (J.L.); ecarino@umd.edu (E.C.); sbera@umd.edu (S.B.); 2Silvec Biologics, Rockville, MD 20850, USA; fgao@silvec.com; 3Department of Cell and Molecular Biology and Biochemistry, University of Missouri-Kansas City, Kansas City, MO 64110, USA; jpmay@umkc.edu

**Keywords:** umbravirus, satellite RNAs, coat-dependent RNA replicon, nonsense-mediated decay, genome-length RNA structure

## Abstract

We report the biological and structural characterization of umbravirus-like associated RNAs (ulaRNAs), a new category of coat-protein dependent subviral RNA replicons that infect plants. These RNAs encode an RNA-dependent RNA polymerase (RdRp) following a −1 ribosomal frameshift event, are 2.7–4.6 kb in length, and are related to umbraviruses, unlike similar RNA replicons that are related to tombusviruses. Three classes of ulaRNAs are proposed, with citrus yellow vein associated virus (CYVaV) placed in Class 2. With the exception of CYVaV, Class 2 and Class 3 ulaRNAs encode an additional open reading frame (ORF) with movement protein-like motifs made possible by additional sequences just past the RdRp termination codon. The full-length secondary structure of CYVaV was determined using Selective 2’ Hydroxyl Acylation analyzed by Primer Extension (SHAPE) structure probing and phylogenic comparisons, which was used as a template for determining the putative structures of the other Class 2 ulaRNAs, revealing a number of distinctive structural features. The ribosome recoding sites of nearly all ulaRNAs, which differ significantly from those of umbraviruses, may exist in two conformations and are highly efficient. The 3′ regions of Class 2 and Class 3 ulaRNAs have structural elements similar to those of nearly all umbraviruses, and all Class 2 ulaRNAs have a unique, conserved 3′ cap-independent translation enhancer. CYVaV replicates independently in protoplasts, demonstrating that the reported sequence is full-length. Additionally, CYVaV contains a sequence in its 3′ UTR that confers protection to nonsense mediated decay (NMD), thus likely obviating the need for umbravirus ORF3, a known suppressor of NMD. This initial characterization lays down a road map for future investigations into these novel virus-like RNAs.

## 1. Introduction

Plant RNA viruses are frequently found in association with subviral RNAs that either share or do not share any sequence similarity with their partner RNA. All subviral RNAs are characterized by their inability to independently complete an infection cycle, which includes transmission to the next host plant. Subviral RNAs range substantially in size and complexity, from non-coding satellite RNAs (194 nt to about 500 nt) that depend on their helper virus for replication, movement and encapsidation [1] to a variety of translated RNAs such as satellite viruses that encode their own capsid protein but depend on the helper virus for replication and movement [2]. Subviral RNAs can markedly affect disease symptoms associated with their helper virus, either attenuating or exacerbating symptoms, which can vary depending on the host. Most subviral RNAs likely exist in a mutualistic association with their helper virus by supporting the helper’s infection cycle. For example, satC associated with turnip crinkle virus (TCV; betacarmovirus), restricts virion formation, thus providing additional monomeric capsid proteins for suppressing RNA silencing [3,4] and/or additional unencapsidated TCV genomic (g)RNA for systemic spread [5,6].

A more recent grouping of coding subviral RNAs is the coat protein-dependent RNA replicons, which encode their own RNA-dependent RNA polymerase (RdRp) but are dependent on a helper virus for at least encapsidation [7]. Umbraviruses are the most studied members of this group, and are always found associated with a helper virus from one of the three genera of the phloem-restricted family Luteoviridae (luteoviruses, poleroviruses and enamoviruses) [8]. Umbraviruses, such as pea enation mosaic virus 2 (PEMV2) (Figure 1A) are classified as members of the Tombusviridae based on having similar RdRp sequences. All umbraviruses encode a replication-required protein (ORF1) and their RdRp (ORF2), which is synthesized by an infrequent −1 ribosomal frame-shifting (-1PRF) event that occurs just upstream from the ORF1 termination codon [9]. Umbraviruses also encode two movement proteins from overlapping ORFs that are expressed from a subgenomic (sg)RNA: ORF3 is the long-distance movement protein and ORF4 is the cell-to-cell movement protein [8,10]. The mutualistic association between umbraviruses and their helper viruses is demonstrated by helper virus acquisition of cell-to-cell movement through shared use of the ORF4 umbravirus product, and umbraviruses gaining a suppressor of RNA silencing and genome encapsidation from their associated virus. In addition, since the umbravirus ORF3 product is multifunctional and also responsible for protecting the umbravirus sgRNA (and many cellular mRNAs) from nonsense mediated decay (NMD) [11], the helper virus likely benefits from this protection as well.

A recently recognized type of coat-protein dependent replicon is the tombusvirus-like associated RNAs (tlaRNA) [7]. These intriguing 3 kb subviral RNAs contain only two of the normal 5 tombusvirus ORFs (ORF1 and ORF2). Like tombusviruses, the RdRp encoded by ORF2 is generated by an infrequent ribosome read-through event following translation of ORF1. tlaRNAs are able to replicate independently in single cells, but are dependent upon their helper polerovirus for systemic movement and trans-encapsidation [7,12]. Besides efforts to understand the phylogenetic relationships among tlaRNAs [7], little is known about any RNA structural features or biology. All tlaRNAs produce a similarly sized, highly abundant non-coding sgRNA, which is co-terminal with the 3′ end of the tlaRNA gRNA and thus (based on size) should be composed of only 3′ UTR sequences. The gRNA and sgRNA of tlaRNAs accumulate to higher levels in the presence of a silencing suppressor expressed in transgenic host plants, suggesting that tlaRNAs benefit from the silencing suppressor activity supplied by their helper virus [7]. In reciprocation, tlaRNAs enhance the accumulation of their helper virus in coinfected cells by a mechanism that is not yet understood [12,13,14].

Another group of coat protein-dependent replicons are similar to tlaRNAs in their coding capacity (ORFs 1 and 2) but their RdRps are reported to be more closely related to those of umbraviruses (Figure 1). As with umbraviruses, these umbravirus-like associated RNAs (ulaRNAs) generate their RdRp by a -1PRF event [15,16,17]. ulaRNAs infecting papaya and babaco in Brazil, Ecuador and Mexico are larger in size than tlaRNAs, with the size difference mainly due to the expanded length of their 3′ UTRs. ulaRNA papaya meleira virus 2 (PMeV2) is 4.5 kb in length and is exclusively associated with helper virus papaya meleira virus (PMeV), a dsRNA virus related to totiviruses [16,18]. Together, PMeV and PMeV2 are responsible for papaya sticky disease, a major agricultural problem in Brazil and Mexico that leads to complete crop devastation. A similar large ulaRNA (4.6 kb) found in 23% of nursery babaco plants and 68% of orchard babaco plants does not generate detectable symptoms and has been named babaco virus Q (BabVQ) [19]. A smaller ulaRNA known as papaya virus Q or papaya umbravirus (PUV; 3.5 kb) was discovered in papaya plants in Ecuador, where it was nearly always found in symptomatic papaya plants associated with the papaya ringspot virus (potyvirus) [15,18]. Although umbraviruses are exclusively associated with viruses in the family Luteoviridae, and despite efforts to identify a member of the Luteoviridae in infected papaya plants, no such helper viruses have been identified [15,16,17].

**Figure 1 viruses-13-00646-f001:**
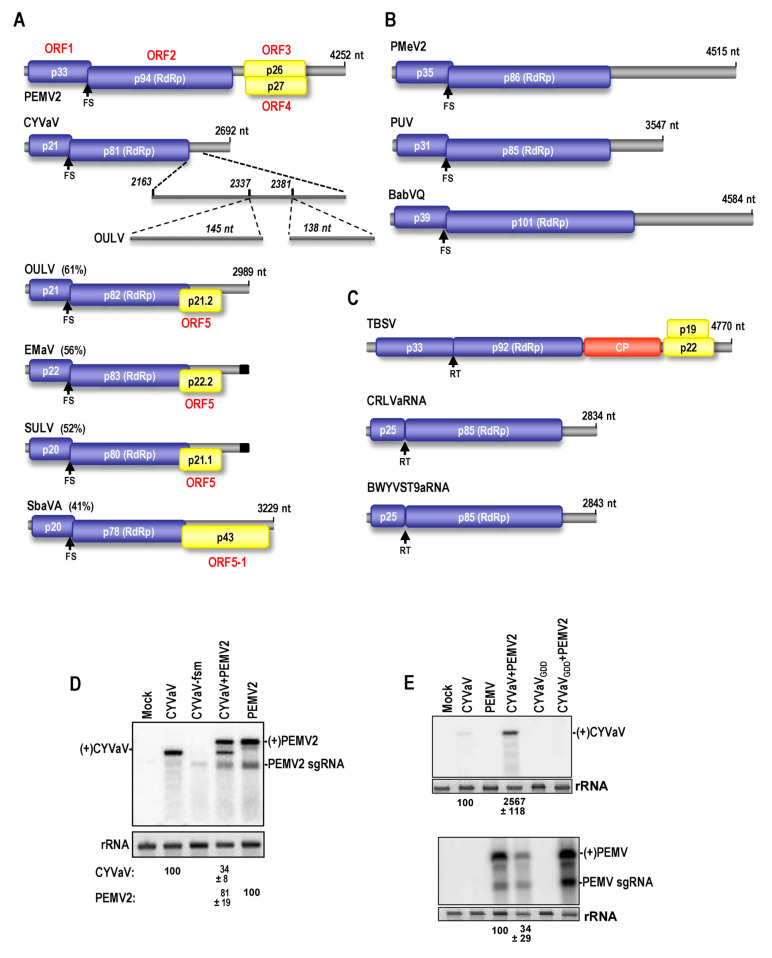
Gene organization of ulaRNAs and related viruses. (**A**) Genomes of the smaller ulaRNAs and umbravirus PEMV2. ORF1 and the -1PRF product (the RdRp) are found in all umbraviruses and ulaRNAs. The umbravirus ORF3 product is the long-distance movement protein and suppressor of NMD whereas the ORF4 product is required for cell-to-cell movement. ORF5, found in OULV, EMaV and SULV, and ORF5-1, found only in SbaVA, code for proteins with movement protein motifs (see Appendix A) and are possibly translated from an sgRNA (see Figure 9). CYVaV differs from OULV, EMaV and SULV by the absence of two fragments, which for OULV are 145 and 138 nt. The current length of SULV and EMaV are unknown, due to issues at the 5′ and 3′ ends (see text), which is represented by black boxes at the 3′ end of their genomes. Percentages shown denote sequence identify with CYVaV. FS, -1PRF site. (**B**) Papaya and babaco ulaRNAs. (**C**) Genome organization of tomato bushy stunt virus (TBSV; tombusvirus) and two tlaRNAs, carrot red leaf virus associated RNA (CRLVaRNA) and beet western yellows virus ST9 strain (ST9aRNA). (**D**) Accumulation of CYVaV in protoplasts. *A. thaliana* protoplasts were inoculated with transcripts of CYVaV alone or coinoculated with transcripts of PEMV2 gRNA and total RNA was extracted 40 h later and examined by Northern blots. Blots were probed with combined oligonucleotide probes for both (+)CYVaV and (+)PEMV2. CYVaV, wild-type CYVaV; CYVaV-fsm, CYVaV with two nt alteration in the frameshift slippery site that suppresses translation of p81 (see Figure 8). Ethidium bromide-stained 28S rRNA was used as a loading control. Quantitation is from three independent experiments and standard deviation is shown. (**E**) Northern blot of RNA extracted from *N. benthamiana* leaves 7-days post-infiltration with CYVaV and/or PEMV2. Upper and lower panels were probed for (+)CYVaV and (+)PEMV2, respectfully. CYVaV_GDD_, CYVaV with a polymerase-inactivating mutation in the GDD active site. Experiments were repeated at least three times and average effect of the two gRNAs on each other′s accumulation is shown along with standard deviation.

A second class of ulaRNAs was first isolated in *Opuntia ficus indica* fruit cactus plants with symptoms of pad swelling disease [17]. Although described in the report as an umbravirus, the opuntia ulaRNA (known as opuntia umbravirus-like RNA; OULV) does not contain umbravirus ORF3 and ORF4. The gene organization and aa sequences of OULV ORF 1 and 2 products are similar to umbraviruses and the papaya/babaco ulaRNAs. In contrast with these larger ulaRNAs, at least one additional ORF was postulated for OULV that overlaps with the end of ORF2 in the −1 frame. Although a virus with sequence similarity to poleroviruses was found in aphids feeding on the opuntia plants, no evidence for this virus was found in opuntia containing OULV [17]. Similar ulaRNAs sequences from sugarcane (sugarcane umbra-like virus; SULV) [20] and citrus (citrus yellow vein associated virus; CYVaV) (Kwon et al., submitted) have been reported, and additional sequences from Ethiopian maize (Ethiopian maize associated virus; EMaV) and strawberry (strawberry associated virus A; SbaVA) have been deposited in GenBank (Figure 1).

For this report, we conducted the first genome-wide biological and structural characterization of ulaRNAs. OULV, SULV and EMaV differ from CYVaV by containing one additional ORF (ORF5) that contains motifs from the 30 kDa class of movement proteins. ORF5 overlaps with the end of ORF2 and is made possible by (at least) the presence of two similar length sequences at the same locations downstream of ORF2 that are absent in CYVaV. The complete secondary structure of CYVaV was determined using Selective 2’ Hydroxyl Acylation analyzed by Primer Extension (SHAPE) structure probing and was employed as a template for determining putative full-length structures for OULV, SULV and EMaV, and local RNA structures for the other ulaRNAs. The smaller ulaRNAs have 3′ terminal elements similar to those of nearly all umbraviruses, which differ from the 3′ terminal elements of tlaRNAs, which resemble those of tombusviruses. The ribosome recoding sites of the smaller ulaRNAs, as well as PUV and BabVQ, differ markedly from those of umbraviruses and PMeV2, and may exist in two conformations. Remarkably, the CYVaV genome was 6-fold more efficient at ribosomal frame-shifting in vitro than umbravirus PEMV2. Unlike PEMV2, CYVaV and OULV replicated independently in protoplasts in the absence of an ORF3 umbravirus product, with CYVaV protected from NMD by an unknown element within its 3′ UTR. Unlike tlaRNAs, no prominent sgRNA was found associated with CYVaV and OULV despite the presence of a conserved umbravirus sgRNA start site located just upstream of ORF5 that is absent in ORF5-deficient CYVaV. Phylogenetic analyses suggest that (i) ulaRNA RdRp nucleotide sequences are more conserved than the aa sequences; (ii) CYVaV has lost two segments found in the most related ulaRNAs that are required to generate ORF5 and (iii) umbraviruses evolved from a common ancestor with the insertion of ORFs 3 and 4. This initial characterization of ulaRNAs lays down a road map for future investigations into these novel virus-like RNAs.

## 2. Materials and Methods

### 2.1. Constructs Used in This Study

The pET17b-CYVaV infectious clone containing full-length CYVaV gRNA cDNA (GenBank accession number JX101610) [21] was developed and provided by the Vi-dalakis laboratory at the University of California, Riverside. The CYVaV cDNA was then cloned into pUC19 downstream of a T7 RNA polymerase promoter by liga-tion-independent cloning [22], generating pUC-CYVaV. Plasmids were linearized with Hind III and used as templates for in vitro transcription using T7 RNA polymerase, which generates uncapped RNA transcripts with the exact 5′ end and one non-template residue at the 3′ end. For agroinfiltration, pCB-CYVaV was generated containing full-length CYVaV cloned into binary plasmid pCB301 [23] downstream of the CaMV 35S promoter and upstream of the hepatitis delta virus ribozyme to generate authentic 5′ and 3′ ends when transcribed. CYVaV mutants were generated using DNA oligonucleotides (IDT) and one-step site-directed mutagenesis [24]. All mutations were verified by sequencing. pCB-PEMV2 (pCB301 harboring full-length PEMV2 gRNA) was previously described [11]. GFP-3′UTR_PEMV_ (the 3′ UTR of PEMV2 immediately downstream of the GFP ORF in pBIN61S-GFP) was also previously described [11,25]. GFP-3′UTR_CYVaV_ was constructed by ligating a DNA fragment encompassing the CYVaV 3′ UTR into the BamHI and XbaI restriction sites of pBIN61S-GFP.

### 2.2. Inoculation of Protoplasts and RNA Extraction

Protoplasts were prepared from callus cultures of *Arabidopsis thaliana* (ecotype Col-0) as described previously [26]. Protoplasts (4 × 10^5^) were transfected with 4 µg of in vitro synthesized RNA transcripts using a polyethylene glycol-mediated transformation protocol. Cells were collected at the desired time point and total RNA was prepared using RNA extraction buffer (50 mM Tris-HCl (pH 7.5), 5 mM EDTA (pH 8.0), 100 mM NaCl and 1% SDS), followed by phenol-chloroform extraction and ethanol precipitation.

### 2.3. Agrobacterium tumefaciens Infiltration and RNA Extraction

*Agrobacterium tumefaciens* was renamed *Rhizobium radiobacter* but the original name will be used here for clarity. Agrobacteria infiltration (agroinfiltration) was performed as previously described [4]. Agrobacterium strains C58C1 (initial experiments only) or GV3101 carrying individual constructs were cultured in LB medium supplemented with appropriate antibiotics at 200 rpm at 28 °C overnight, and then diluted 500-fold with fresh LB medium and incubated under the same conditions overnight. Agrobacteria were precipitated (2800 rpm, 4 °C, 10 min) and then resuspended in infiltration buffer (10 mM MES (pH 5.6), 10 mM MgCl_2_ and 100 μM acetosyringone) to an OD_600_ of 1.0, and then incubated at room temperature for at least 2 h prior to infiltration, which was carried out using a 1 mL needleless syringe on the first two true leaves of three-week-old *Nicotiana benthamiana* (containing 4–5 leaves). Agrobacteria containing binary vectors with full-length CYVaV or PEMV2 genomes were coinfiltrated with the p14 silencing suppressor of pothos latent virus [27] at an OD_600_ ratio of 0.4:0.2. Total RNA from *N. benthamiana* infiltrated leaves was isolated using TRIzol reagent (Invitrogen) according to the manufacturer’s protocol.

### 2.4. Northern Blots

Total RNA was subjected to agarose gel electrophoresis and blotting, and membranes were hybridized with [γ-^32^P] ATP-labeled oligonucleotides. Plus-strand CYVaV ((+)CYVaV) probes were complementary to positions 2202–2239, 2464–2502 and 2655–2692. Minus-strand CYVaV ((−)CYVaV) probes were (+)-strand positions 325–342, 855–873, 1375–1392, 1895–1912 and 2416–2434. (+)PEMV2 probes were 2969–3004, 2731–2771 and 3229–3270. (+)OULV probes were complementary to positions 2950–2989, 2757–2796 and 2505–2554. (−)OULV probes were complementary to (+)-strand positions 404–423, 914–935, 1432–1452, 2099–2123 and 2597–2622.

### 2.5. In Vitro Translation

Plasmids containing full-length CYVaV (pUC19-CYVaV) or PEMV2 (pUC19-PEMV2) were linearized with Hind III or SmaI, respectively, and served as templates for RNA transcription using T7 RNA polymerase. Quantification of the synthesized RNA was performed using a DeNovix DS-II FX spectrophotometer. For in vitro translation, 10 μL translation mixtures contained 5 μL wheat germ extract (Promega), 0.5 pmol RNA template, 0.8 μL 1 mM amino acids mix (without methione), 100 mM potassium acetate and 0.5 μL (5 μCi) ^35^S-methionine. The translation mixture was incubated at 25 °C for 45 min and then resolved on a 10% SDS-PAGE gel. The gel was dried and exposed to Fuji phosphorimager screen for 3 h. The screen was subsequently scanned by an Amersham Typhoon fluorescent image analyzer. The intensity of radioactive bands was quantified using ImageQuant TL 8.1 (GE Lifesciences). All experiments were repeated at least three times.

### 2.6. Preparation of RNA Template for SHAPE Probing

pUC19-CYVaV was linearized using HindIII and linearized plasmid treated with phenol:chloroform followed by 3 M sodium acetate precipitation. Linearized pUC19-CYVaV was used to synthesized full length RNA via T7 polymerase in vitro transcription; briefly, 5 µg of linearized pUC19-CYVaV was incubated for 2 h in a 100 µL in vitro transcription reaction composed of 6 mM DTT, 0.6 mM rXTPs, 24 mM Tris pH 8, 15 mM NaCl, 4.8 mM MgCl_2_, 1.2 mM spermidine, 1 U/µL RNase inhibitor (NEB) and 3 U/µL of T7 RNA polymerase. Synthesized RNA was treated with 1 U of RNAse-free DNAse (Promega) to remove the DNA template. DNase was removed by treatment with phenol:chloroform followed by lithium chloride precipitation of the RNA. RNA was quantified using a DeNovix DS-II FX spectrophotometer and the quality of the RNA was checked in a 1% TBE native agarose gel.

### 2.7. SHAPE RNA Structure Probing

Nine picomoles of RNA were diluted in 30 µL of water and denatured at 65 °C for 4 min followed by snap cooling on ice for 2 min. RNA was subjected to folding by adding 7.5 µL of 5X RNA folding buffer (80 mM Tris-HCl pH 8.0, 160 mM NH_4_Cl and 11 mM MgOAc) and incubated at 37 °C for 20 min. The folded RNA was divided into two tubes (18.75 µL) with one treated with 3.3 µL of 100 mM N-methylisatoic anhydride (NMIA) in dimethyl sulfoxide (DMSO), and the other treated with DMSO only, for positive or negative samples respectively. The RNA samples were incubated at 37 °C for 45 min to allow interaction with the modifying agent. After modification, the RNA was precipitated with 0.1 volume of 3 M sodium acetate and 2.5 volumes of cold 95% ethanol. The RNA pellet was reconstituted in 0.5X TE buffer (10 mM Tris-HCl pH 8.0 and 1 mM EDTA pH 8.0). RNA was quantified using a DeNovix DS-II FX spectrophotometer and the quality of the RNA was checked in a 1% TBE native agarose gel.

SHAPE-modified RNA was subjected to cDNA synthesis using SuperScript III Reverse Transcriptase (Invitrogen) following the manufacturer’s recommendations. Briefly, 1 pmol of control or modified RNA was diluted in 8 µL of 0.5X TE buffer and 4 µL of 2.5 µM 5′ end labeled fluorescent primer (6FAM was used for SHAPE modified samples and PET was used for sequencing ladder samples). Tubes were incubated at 65 °C for 5 min followed by ice incubation for 5 min. Next, 8.5 µL of SHAPE enzyme mix (1X SSIII FS buffer (250 mM Tris-HCl (pH 8.3), 375 mM KCl, 15 mM MgCl_2_), 5 mM DTT, 0.5 mM dNTPs, 1U SSIII enzyme) or sequencing ladder mix (1X SSIII FS buffer, 5 mM DTT, 0.5 mM dNTPs, 0.1 µmol of ddTTP and 1U SSIII enzyme) was added to the corresponding samples, which were incubated at 52 °C for 1 h. cDNA samples were treated with 1 µL of 4 M NaOH and heated at 95 °C for 5 min to degrade the RNA. Samples were then neutralized by adding 2 µL of 2 M HCl. Prior to cDNA precipitation, half of the sequencing ladder cDNA reaction (PET) was added to the corresponding SHAPE-modified (6FAM) cDNA sample ((−) DMSO or (+) NMIA/DMSO). The cDNA reaction mixtures were precipitated with 0.1 volumes of 3 M sodium acetate and two volumes of cold >95% ethanol. cDNA pellets were washed with 70% ethanol, air dried and resuspended in 15 µL of nuclease free water. The samples were then provided to Genewiz for fragment analysis. The cDNA fragmentation data obtained from Genewiz was scanned through Peak Scanner 2 (Invitrogen) to check for the presence of cDNA fragments in the data sets, and samples were analyzed using QuSHAPE software [28]. Manual adjustments were performed in QuSHAPE to ensure that the program was subtracting background from the corresponding peak positions. Excel was used to align and overlap the QuSHAPE data for each primer set used. Each primer set was repeated 2–3 times and the average reactivity data served as input for RNA structure drawing program RNA2Drawer [29], using the on-line version (https://rna2drawer.app/). To determine the secondary structure, mFOLD [30] was used to create a base structure, which was modified based on phylogenetic analysis of conserved structures in CYVaV and Class 2 ulaRNAs using the pair complement tool in RNA2Drawer [29] and SHAPE reactivity data. Primers used for SHAPE probing were: CY297–318: 5′-CCCAACCAACTTCTACGATATC, CY783–805: 5′-AGAAGAGGGAAGGATCGGATCTC, CY1046–1067: 5′-GCTTCCTTGGTTGGATTGCTCG, CY1388–1412: 5′-ACTCATATGACAATTCTTTGCAGGA, CY1563–1584: 5′-GGGCTGGGGACGTCGTGAGACT, CY1763–1784: 5′-TATATTCAATCGTGGAGCCCGT, CY1898–1920: 5′-CCTGATTTGAAGAAAACCCCTCG, CY2066–2087: 5′-TAGTTGTTTCTATTTTACTCAG, CY2272–2296: 5′-ACACCTACATTAACTGGTTAGGTT, CY2663–2680: 5′-CTCTCCCCACGTGACAC.

### 2.8. Transient NMD Assay and RT-qPCR

GFP reporters were agroinfiltrated with the p14 silencing suppressor and either mock or the dominant-negative Upf1 inhibitor of NMD as previously described [25]. At 5 days post-infiltration, leaves were imaged and collected for RNA extraction using Trizol. Total RNA was treated with RQ1 DNase (Promega) prior to reverse-transcription quantitative PCR (RT-qPCR). Small PCR fragments (<200 bp) were amplified using the SYBR Green-based Luna One-Step RT-qPCR kit according to the manufacturer’s protocol (New England BioLabs). The Roche LightCycler 480 platform was used for all experiments. p14 served as an internal reference gene for relative GFP gene expression using the 2^−∆∆Ct^ method.

## 3. Results and Discussion

### 3.1. Genome Organization of ulaRNAs

Genome organization of viral and virus-like RNAs found in this report are shown in Figure 1A–C. Similar to PMeV2, BabVQ and PUV (Figure 1B), CYVaV (2692 nt) contains only two ORFs (Figure 1A). Based on numerous reports of viruses in the family Tombusviridae, CYVaV ORFs1 and 2 encode a 21 kDa replication-required product and an 81 kDa extension product with the hallmark GDD motif of polymerases. OULV, EMaV, SULV and SbaVA share 61%, 56% and 52% and 41% overall nucleotide sequence identity, respectively, with CYVaV. Compared with CYVaV, OULV, EMaV and SULV had two additional segments of similar lengths in identical locations past the ORF2 termination codon. These extra sequences contribute to the presence of an additional ORF, denoted ORF5 (to distinguish it from related umbravirus and ulaRNA ORFs 1 and 2, and umbravirus-specific ORFs 3 and 4). SbaVA also contains an additional ORF, which encodes a substantially larger protein and thus is designated ORF5-1. Alignment of ORF5/ORF5-1 aa sequences revealed that the ORF5 encoded product contains conserved motifs found in the 30 kDa class of viral movement proteins [31], which are mostly absent in the ORF5-1 product of SbaVA, suggesting that at least the shorter ORF5 products may be associated with RNA movement (Appendix A). The alignment also suggests that ORF5-1 of SbaVA may have a different origin from ORF5 of OULV, EMaV and SULV. All ulaRNAs are similar to umbraviruses in generating their RdRp by a -1PRF event upstream from the termination codon of ORF1, which differs from tlaRNAs, which generate their RdRp by a ribosomal read-through event, similar to tombusviruses (Figure 1C).

### 3.2. ulaRNA Replication in Single Cells

To ensure that the CYVaV sequence reported in GenBank represented the complete full-length gRNA, an expression plasmid containing the reported sequence was generated to determine if CYVaV transcripts were capable of replication. *Arabidopsis thaliana* protoplasts were inoculated with in vitro synthesized CYVaV transcripts containing the exact 5′ end and one non-template residue at the 3′ end, and total RNA was isolated 24 h later. Since efficient accumulation of PEMV2 in protoplasts is dependent on the presence of the ORF3 product p26 [11,32], which is absent in CYVaV, cells were also inoculated separately or together with PEMV2 RNA, and probes complementary to CYVaV and PEMV2 plus (+)strands were combined and used for Northern blot analysis. As a control, CYVaV transcripts containing mutations in sequence required for frame-shifting (CYVaV-fsm; shown in Figure 8) were also assayed for accumulation. As shown in Figure 1D, full-length wild-type CYVaV was capable of independent replication and levels were reduced, not enhanced, by the presence of PEMV2. In contrast, PEMV2 levels were not significantly affected by the presence of CYVaV.

Different results were obtained when CYVaV and PEMV2 were agroinfiltrated into leaves of young *N. benthamiana* plants (Figure 1E). Agroinfiltration leads to synchronous accumulation of replication-positive, genome-length plus (+)-strand transcripts generated from a plasmid containing the cauliflower mosaic virus 35S promoter. In infiltrated leaves, PEMV2 levels decreased 2.9-fold when coinfiltrated with CYVaV, and CYVaV levels increased by 25-fold in the presence of PEMV2. The increase in CYVaV accumulation was not due to trans-replication by the PEMV2 RdRp, as no accumulation of CYVaV was detected when PEMV2 was coinfiltrated with CYVaV containing a mutation in its RdRp active site (CYVaV_GDD_). One possibility is that CYVaV was benefiting in *N. benthamiana* plants, but not *Arabidopsis* protoplasts, from the presence of p26. No CYVaV sgRNA was discernable in either protoplasts or infiltrated leaves, which differs from tlaRNAs, which all have a prominent sgRNA [7,33].

### 3.3. Taxonomic Relationships of ulaRNAs

To begin resolving the taxonomic position of ulaRNAs, we generated a phylogenetic tree based on the complete RdRp aa sequences from ulaRNAs, tlaRNAs, umbraviruses and selected tombusviruses and betacarmoviruses. Initial trees generated using barley yellow dwarf virus (BYDV; luteovirus) as an outgroup failed (BYDV was not an outgroup on the tree) and thus, the tree was mid-point rooted. The aligned viral sequences were checked for recombinants using the RDP4 package [34], which revealed that tlaRNA cucurbit aphid borne associated RNA and pelargonium leaf curl virus (tombusvirus) might be recombinants and thus they were removed from subsequent analysis. As shown in Figure 2A, the larger papaya and babaco ulaRNAs comprise one class (Class 1), CYVaV, OULV, EMaV, SULV form a second class (Class 2) and SbaVA, which is in a sister clade to Class 1, comprises a third class (Class 3). Class 1, Class 2 and Class 3 ulaRNAs cluster together in a monophyletic clade with high bootstrap support (100%) and all classes of ulaRNAs occupied a sister clade position relative to umbraviruses (high support of 94.3%). The tree also indicates strong support (94.3%) for a monophyletic clade of tlaRNAs with tombusviruses and betacarmoviruses, which all use ribosomal read-through to generate their RdRp. This tree differs from a similar tree generated using only conserved RdRp motifs in ORF2, which places the tlaRNAs closer to the ulaRNAs (Kwon et al., manuscript submitted).

The RdRp region of ulaRNAs consists of conserved structural RNA elements (see below), and thus the RNA sequence is not only acting as an ORF. Therefore, a second phylogenetic tree was developed based on the full-length nucleotide sequence of the RdRp ORF (Figure 2B). Similar to the tree built on the RdRp aa sequence, Class 1, Class 2 and Class 3 ulaRNAs clustered together with a high support of 100%. Although both trees had similar topologies, the nucleotide substitutions per site (0.2; Figure 2B) was less than the aa substitutions per site (0.4; Figure 2A), suggesting that among the analyzed viruses and virus-like RNAs, the nucleotide sequences in the RdRp ORFs were more closely related than the encoded protein aa sequences.

**Figure 2 viruses-13-00646-f002:**
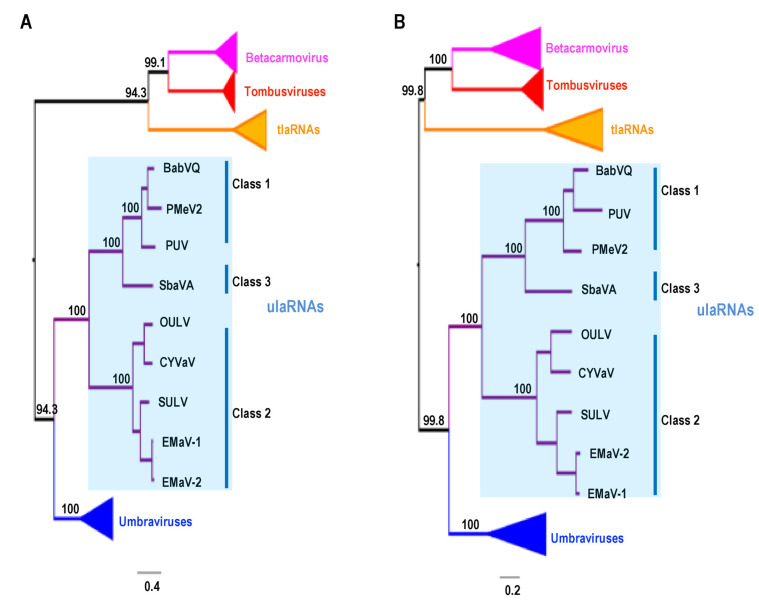
Maximum likelihood phylogenetic tree based on the amino acid (**A**) and nucleotide (**B**) sequence of RdRp from ulaRNAs, 6 tlaRNAs and 24 viruses from the umbravirus, tombusvirus and betacarmovirus genera. Branch numbers indicate bootstrap support in percentage out of 1000 replicates. The scale bar denotes nucleotide/protein substitutions per site. Both trees were mid-point rooted. BabVQ: babaco virus Q (MN648673); CMoV: carrot mottle virus (FJ188473); CMoMV: carrot mottle mimic virus (U57305); CYVaV: citrus yellow vein associated virus (JX101610) [21]; EMaV-1 and EMaV-2: Ethiopia maize-associated virus (MN715238, MF415880); ETBTV: Ethiopian tobacco bushy top virus (KJ918748); GRV: groundnut rosette virus (MG646923); IxYaV2: ixeridium yellow mottle associated virus 2 (KT946712); OPMV: opium poppy mosaic virus (EU151723); OULV: opuntia umbra-like virus (MH579715); PMeV2: papaya meleira virus 2 (KT921785); PUV: papaya umbra virus (KP165407); PMMoV: patrinia mild mottle virus (MH922775); PEMV2: pea enation mosaic virus 2 (U03563); RCUV: red clover umbravirus (MG596237, MG596235); SULV: sugarcane umbra-like virus (MN868593); strawberry associated virus A (MK211274); TBTV: tobacco bushy top virus (KX216407).

### 3.4. Full-Length Secondary Structure of CYVaV

When the results of the two trees are considered together, they imply that the RdRp region of ulaRNAs might be undergoing differential selection pressures; one at the protein level and a second at the RNA level to maintain functional RNA structures. RNA structures throughout RNA virus genomes are engaged in functions important to the virus infection cycles. For umbraviruses and related betacarmoviruses and tombusviruses, distinctive *cis*-acting structures located at or near the 5′ and 3′ ends promote replication and translation [35,36,37,38,39,40,41,42] and structures in internal locations promote ribosome recoding [9,43], sgRNA synthesis [44,45,46,47,48] and binding and organization of the RdRp [37,49]. Unstructured RNA segments also contribute important functions, such as overcoming NMD in betacarmovirus TCV [50] and as internal ribosome entry sites in the same virus [41]. Long-distance base pairing of RNA sequences additionally contribute to the structural and functional organization of RNA genomes, either as elements in the secondary structure or through tertiary interactions [51,52,53,54,55].

To better understand the evolutionary relationships among ulaRNAs and to start identifying regions in the genomes that are important for function, we used SHAPE RNA structure probing [56,57] combined with phylogenic comparisons and computational algorithms [30] to determine the architecture and secondary structure of the full length CYVaV genome (Figure 3). We then used the CYVaV structure as a framework to examine structural conservation in the other Class 2 ulaRNAs (Figure 4). To simplify description of CYVaV structural features, many of which are conserved in some or all Class 2 ulaRNAs, we subdivided the Class 2 genomes into three domains: Domain 1 (D1; bases 1–668; numbering is from CYVaV) includes the 5′ terminus through the highly conserved structure just past the p21 UGA termination codon and the recoding site structure; Domain 2 (D2; bases 669–2485) is the extended branched hairpin structure that brings together 5′ proximal and 3′ proximal sequences in a long-distance interaction at the base of the hairpin and includes a portion of the 3′ UTR and Domain 3 (D3; bases 2486–2692) comprises the remainder of the 3′ UTR. Each of these domains and their structurally conserved elements are considered separately below.

### 3.5. Domain 1: Overall Structure

D1 contains the short 8 nt 5′ UTR and entire ORF1 of Class 2 ulaRNAs. CYVaV shares 68%, 59% and 58% sequence identify with OULV, EMaV and SULV in D1, respectively, making it the most conserved of the three domains. As shown in Figure 4, all Class 2 ulaRNAs share substantial structural similarities, including six separate stems protruding from the (as drawn) circular backbone (labeled 1–6 in Figure 3 and on the CYVaV structure in Figure 4). Class 3 SbaVA, which shares 43% sequence identity with CYVaV in this region, contains stems 1, 2, 5 and 6 (see below) and the Y-shaped structure that is part of stem 4 (not shown).

### 3.6. Features of Domain 1: The 5′ End

D1 stems 1 and 2 in CYVaV are short hairpins that are strongly supported by the SHAPE probing data. All ulaRNA stem 1′s contain the ultimate or penultimate 5′ nucleotide at the base of the hairpin (Figure 5), with all Class 2 and Class 3 ulaRNAs containing a carmovirus consensus sequence (CCS) at the 5′ end. CCS, which are found at the 5′ ends of nearly all umbravirus gRNAs and at the 5′ ends of all carmovirus and umbravirus sgRNAs, are composed of two or (most commonly) three guanylates followed by a stretch of adenylates and uridylates [58] (in green, Figure 5). For CYVaV, OULV and EMaV, the CCS begins with “GGGUAA” and is just upstream from the ORF1 AUG start codon. The SULV sequence deposited in GenBank does not contain three 5′ terminal guanylates, but does include the adjacent “UAAAU” that is conserved in CYVaV and EMaV, suggesting that the reported sequence may be missing the guanylates at the 5′ end. For EMaV, one of the two reported sequences (MN715238; EMaV-1) contains UUCCGAUCU at the 5′ terminus preceding its CCS (GGUAAAU) while the second (MF415880; EMaV-2; shown in Figure 5) initiates with a CCS (GGGGUAAAU) with one extra guanylate. Additional experimentation is thus needed to determine the correct 5′ terminal sequence for SULV and EMaV.

SULV differs from the other Class 2 ulaRNAs in that the AUG that corresponds with the other class member’s start codons and is in an excellent Kozak context (AAUAUGG) is out of frame with most of ORF1, and ribosomes initiating translation at this AUG would terminate at an UAG stop codon after 41 amino acids. An in-frame AUG in a weaker Kozak context (AGGAUGA) is located at position 64, and this AUG is not present in the other Class 2 ulaRNAs. The exceptional nature of the SULV ORF1 translation initiation site suggests that additional information is needed to confirm its ORF1 initiation codon.

Although the genomes of Class 1 and Class 3 ulaRNAs are too distantly related to CYVaV for genome-length structures to be predicted with any confidence, local regions of structural similarity with Class 2 ulaRNAs were discernable. All of these ulaRNA genomes contain two putative hairpins at their 5′ ends, but only PUV and SbaVA hairpins resemble those of Class 2 ulaRNAs in having a CCS followed closely by the ORF1 initiation codon (Figure 5). In addition, only PUV contains a second hairpin similar to those of Class 2 ulaRNAs. As with SULV, the SbaVA sequence in GenBank does not initiate with guanylates, however their possible presence would significantly strengthen the base of stem 1.

### 3.7. Features of Domain 1: The Recoding Site

The PEMV2 -1PRF site, which is structurally conserved in other umbraviruses, is shown at the bottom left of Figure 6. As with all umbravirus -1PRF sites, it consists of two hairpins: Hairpin A (HA) and a downstream hairpin known as the RSE (recoding stimulatory element) [9]. Umbravirus RSE contain the ORF1 termination codon at their 5′ base, and residues within an asymmetric loop (shaded gray, Figure 6) connect in a long-distance RNA–RNA interaction with the apical loop of the 3′ terminal hairpin [9], similar to comparable interactions throughout members of the family Tombusviridae [9,43,59,60,61]. Pseudoknots within RSE and structural plasticity are also features of some recoding sites in members of the family Tombusviridae and other viruses [43,62,63,64,65].

The site of ribosomal frame-shifting is one of the most conserved regions in Class 2 ulaRNAs and is structurally conserved in all other ulaRNAs with the exception of PMeV2. The RSE can be configured in two conformations: the conformation shown in Figure 6 and Appendix A, which contains the HA hairpin and a two hairpin RSE (H1 and H2) and is poorly supported by the SHAPE data in the base stem of H1, and an alternative conformation shown in Figure 7 and Appendix A that is more consistent with SHAPE. Compensatory mutations generated in the CYVaV RSE H2, however, revealed that this small hairpin is necessary for efficient frame-shifting, supporting the validity of the structure shown in Figure 6 (F. Gao and A.E. Simon, unpublished). The ORF1 termination codons for all ulaRNAs (with the exception of BabVQ and PMeV2) are within HA, and nearly all are 12 nt downstream from a sequence characteristic of ribosome slippery sites (Figure 6, Figure 7 and Figure 8, in orange). The slippery site, where the ribosome switches frames by slipping back one nucleotide, is generally a X_XXY_YYH motif, where XXX are any three identical nucleotides, YYY are typically UUU or AAA and H is A, C or U. In EMaV, a UGA is found at the same position as the stop codons for the other Class 2 ulaRNAs; however it is preceded by an in-frame UAG codon 6 nt upstream.

Sequence in the RSE H1 terminal loop is complementary to two sequences in identical locations in all Class 2 and Class 3 ulaRNAs: one sequence is located near the 3′ terminus (see Figure 12) comparable to locations in members of the Tombusviridae. An additional complementary sequence is located within the RSE between H1 and H2 in all ulaRNAs with the exception of PMeV2 (Figure 6). Although Class 1 and Class 3 ulaRNA RSE regions can fold into HA and the two hairpin RSE conformation, there is little sequence conservation with Class 2 ulaRNAs.

The second conformation for the RSE region is a better fit for the SHAPE data and is also conserved among all ulaRNAs with the exception of PMeV2 (Figure 7, Appendix A). In this putative alternative structure, the base of HA is extended, and the upper portion of RSE H1 along with one or two new hairpins split off from a three-way or four-way junction atop a short stem. In this conformation, the proposed local pseudoknot becomes a kissing-loop interaction with the terminal loop of new hairpin H3. Additional experimental evidence will be needed to validate the two proposed conformations and their necessity for in vivo frame-shifting.

**Figure 6 viruses-13-00646-f006:**
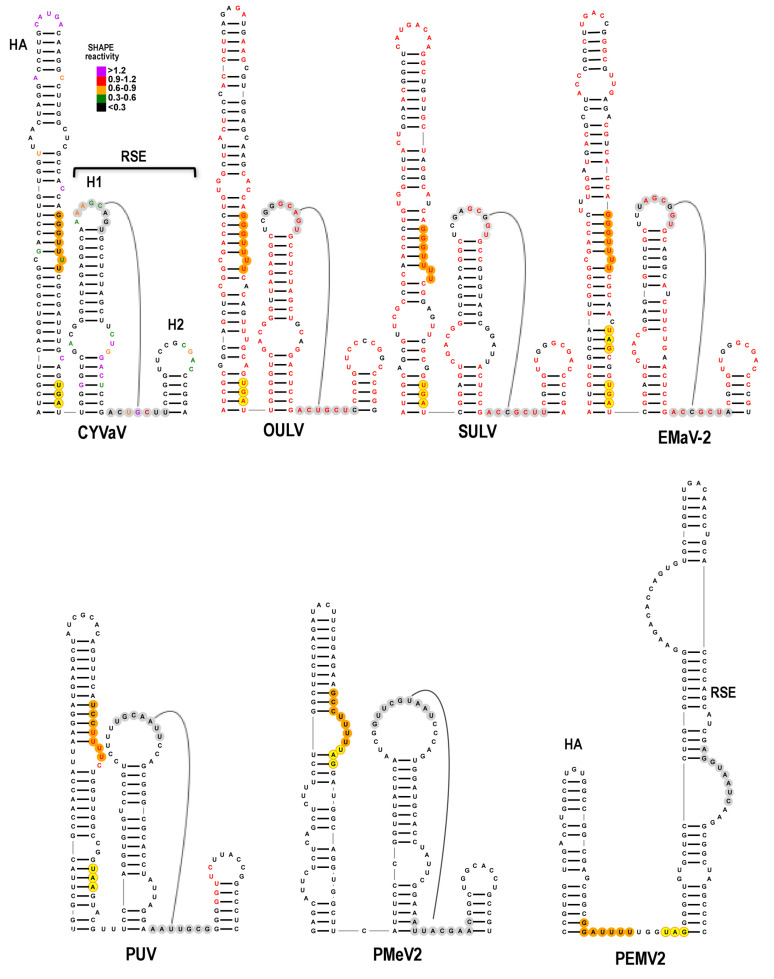
Structures at the -1PRF site in ulaRNAs and umbravirus PEMV2. One of two conformations possible at the recoding site of the ulaRNAs (see Figure 7 for the alternative structure). SHAPE data is presented on the CYVaV structure whereas red residues in the other structures denote sequence conservation with CYVaV. Hairpin A (HA) is a hairpin just upstream from the recoding site, and a hairpin at this location relative to the ribosome recoding site is conserved for all viruses in the Tombusviridae examined. For all Class 2 ulaRNAs with the exception of EMaV, HA contains the slippery site (shaded orange) and ORF1 termination codon (shaded yellow) in identical locations. Note that EMaV has a new upstream in-frame termination codon. Proposed conserved pseudoknots are shown, which would negate the ability of the H1residues to participate in the long-distance interaction with complementary sequence at the 3′ end. See Appendix A for SbaVA and BabVQ structures.

For PUV and SbaVA, the ORF1 stop codons (UAA and UGA, respectively) are similarly located 12 nt downstream from the slippery site within the stem of HA as with most of the Class 2 ulaRNAs. In addition, the PUV, BabVQ and SbaVA sequences, but not the PMeV2 sequence, can adopt a similar alternative conformation but with a three-way junction and two hairpins for PUV and SbaVA (Figure 7; Appendix A). Whereas an alternative structure for the RSE of PMeV2 is discernable, it is much more similar to that of umbravirus PEMV2, with a shorter HA compared with those of Class 2 ulaRNAs, a single extended RSE hairpin, and the slippery site proximal to the UAG stop codon at the base of the RSE.

**Figure 7 viruses-13-00646-f007:**
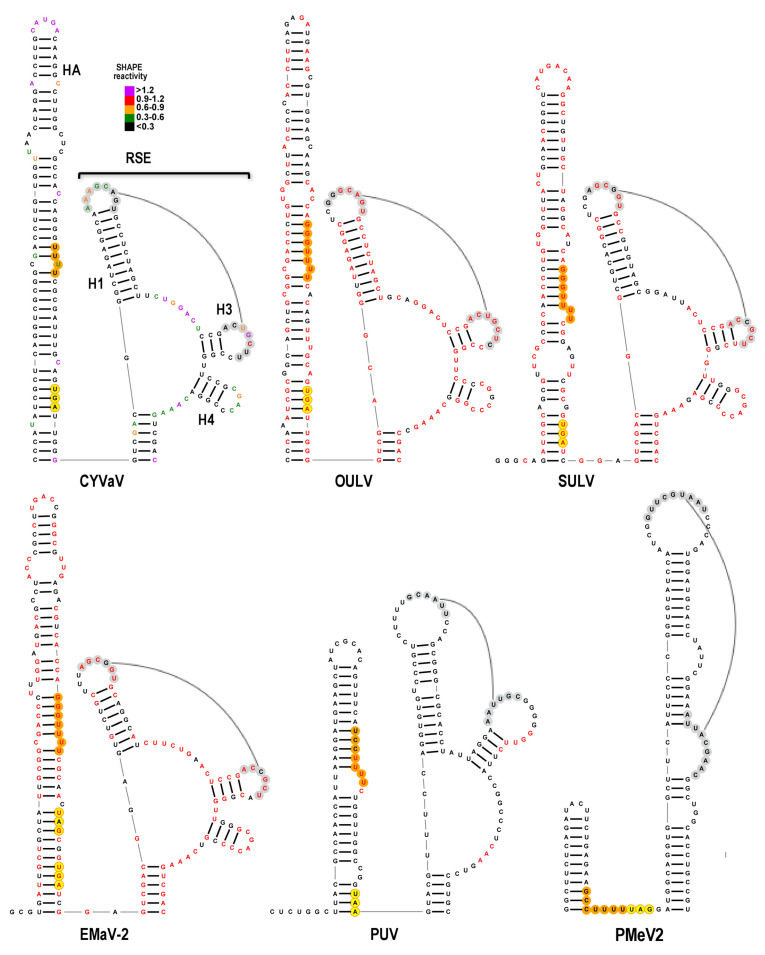
Alternative conformation of structures at the -1PRF site in ulaRNAs. See legend to Figure 6.

Since most of the ulaRNA recoding sites are markedly different from those of umbraviruses, we were interested in determining the efficiency of the frame-shifting event and whether we had correctly identified the slippery site, which is unusually distal from the termination codon and the base of the RSE. Full-length transcripts of CYVaV and PEMV2 were synthesized in vitro and subjected to in vitro translation using wheat germ extracts. The CYVaV recoding site was surprisingly efficient, with a rate of frame-shifting nearly 6-fold higher than that of umbravirus PEMV2 (Figure 8B). Mutations in the proposed slippery site (CYVaV-fsm; Figure 8A) eliminated the p81 frameshift product and insertion of a residue within the p21 termination codon resulted in only p81 synthesis. These results suggest that CYVaV produces a substantially higher amount of RdRp compared with umbraviruses due to one or more of the unusual characteristics of its recoding site.

**Figure 8 viruses-13-00646-f008:**
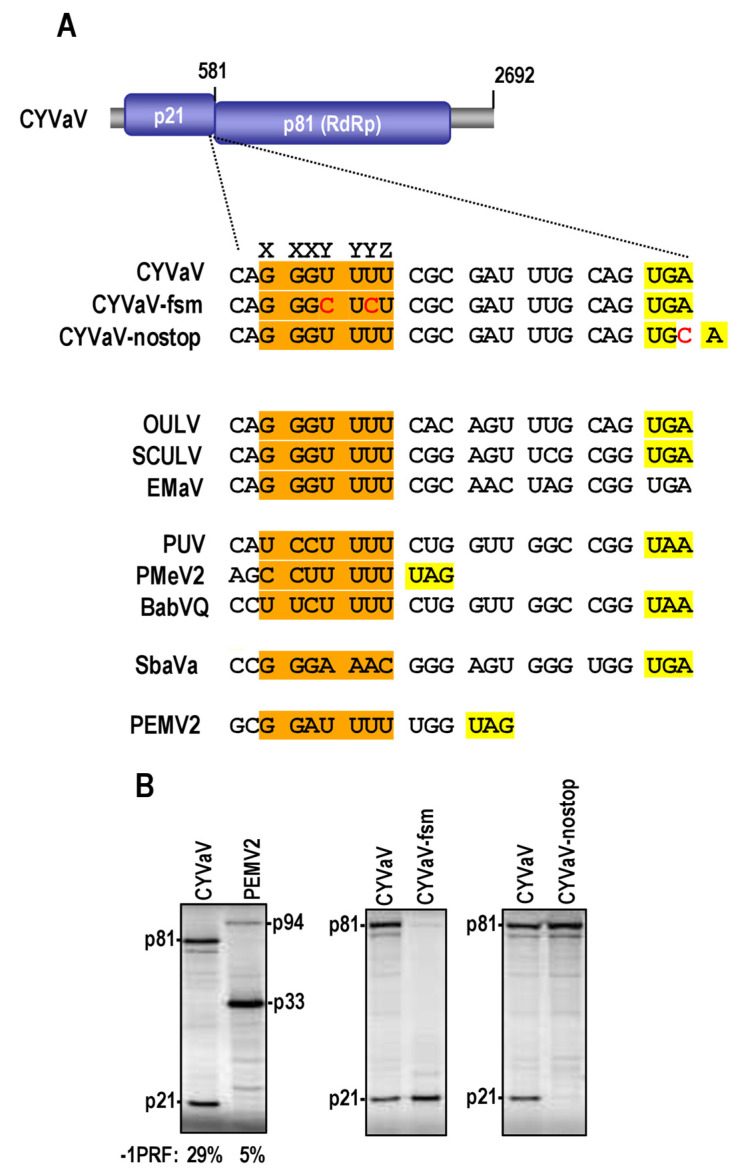
CYVaV recoding site is highly efficient. (**A**) Sequence at the slippery site in ulaRNAs, CYVaV mutants and PEMV2. Proposed slippery sequences are shaded orange. Base alterations are in red. (**B**) In vitro translation of full-length wild-type and mutant CYVaV and wild-type PEMV2 transcripts in wheat germ extracts. Rate of -1 PRF is given.

### 3.8. Domain 2: Overall Structure

D2 contains most of ORF2 and a portion of the 3′ UTR. This domain contains slightly less overall sequence similarity among the Class 2 ulaRNAs, with CYVaV sharing 62%, 55% and 52% identify with OULV, EMaV and SULV, respectively. Although the two inserts in OULV, SULV and EMaV (in red, Figure 4) protrude from conserved stems, their presence and sequence differences are predicted to cause unique structural variations in the 5′ and 3′ regions of this domain. Despite these structural differences, a large number of conserved features are found in Class 2 D2. Conservation of sequence and/or structure in six short segments (red lines in Figure 3) formed the basis for the organization of many of these conserved elements. For example, all Class 2 ulaRNAs contained Structures 7, 9, 12 and 13 (among others not marked), and several others structures were conserved in a subset of Class members (e.g., Structure 8 in CYVaV and OULV, and Structure 11 in EMaV, OULV and SULV). None of these structures were discernable in Class 1 and Class 3 ulaRNAs.

### 3.9. Features of Domain 2: ORF5

As described in Figure 1, OULV, SULV and EMaV differ from CYVaV by the presence of inserts (145 nt and 138 nt for OULV) in similar places in their 3′ UTRs. These two inserts contribute to the existence of ORF5, encoding a protein of 21–22 kDa. Examination of the sequence just upstream from the ORF5 initiation codon in OULV, SULV and EMaV (Structure 11 in Figure 3 and Figure 4) revealed the presence of a typical CCS, GGGUAAAU (Figure 9), which is similar or identical to the 5′ end of the Class 2 gRNAs and suggests that an sgRNA starting with the CCS may be the template for translation of Class 2 ORF5. The RNA structure in the region of the CCS is very similar for these three Class 2 ulaRNAs, and resembles the structure near one of the two sgRNA start sites in umbraviruses (Figure 9) [66]. In the corresponding CCS location in CYVaV, the middle guanylate is replaced by an adenylate, which should eliminate a functional CCS. In addition, the ORF5 AUG initiation codon is replaced by a GUG in CYVaV, and seven termination codons exist in the corresponding reading frame as that of ORF5. CYVaV Structure 11 also differs from that of the other Class 2 ulaRNAs. These observations strongly suggest that CYVaV only encodes ORFs 1 and 2, differing from the other Class 2 ulaRNAs.

tlaRNAs produce high levels of a non-coding sgRNA in infected cells [7,12]. To determine if OULV also produces high levels of an sgRNA in cells that would be the template for translation of ORF5, a plasmid was generated containing the full-length OULV sequence, and in vitro synthesized transcripts were used to inoculate *Arabidopsis* protoplasts. RNA was extracted after 24 h and analyzed by Northern blots. OULV gRNA accumulated to detectable levels in protoplasts, however no discernable sgRNA was present. The variant CCS in CYVaV was also converted to a canonical sequence by replacing the adenylate in position 2 with a guanylate (GAGUAAAAUA to GGGUAAAAUA; construct A2069G). While A2069G was infectious and accumulated to near WT levels in protoplasts (Figure 9E), no sgRNA of the expected size was detected. Although it seems likely that an sgRNA is produced from this CCS in cells infected with OULV, the amount synthesized would be substantially below the levels of tlaRNAs sgRNAs.

For Class 3 SbaVA, ORF5-1 is extended, encoding a protein of 43 kDa. Analysis of the sequence in the vicinity of the initiation codon revealed a putative upstream structure that resembles the two stem-loop Class 2 Structure 11 (Appendix A). The upper portion of the 3′ stem-loop shares significant sequence similarity with those of OULV, EMaV and SULV. However, the Class 2 CCS (GGGUAAAAUUA) was replaced by GAUAAA, and the AUG initiation codon was located 50 nt downstream and not 11 nt downstream of the CCS as found for Class 2. The termination codon for the 43 kDa protein was located only 43 nt from the SbaVA 3′ end, partly accounting for its extended length. The weak similarity in the aa sequence between ORF5-1 and ORF5 (Appendix A), combined with the absence of Class 2-conserved structures in the ORF5-1 RNA sequence (see below), suggests that ORF5 and ORF5-1 may have had different origins.

### 3.10. Features of Domain 2: Conserved Hairpins at the End of ORF2

One conserved feature in all Class 2 ulaRNAs (but not Class 1 or Class 3) is the location of the RdRp termination codon within an extended hairpin with a second long hairpin located just downstream (Structures 12 and 13 in Figure 3 and Figure 4). Since the sequences of these two hairpins share no extensive similarity among Class 2 ulaRNAs (Figure 10C), conservation of a two hairpin structure at this location may be necessary for some function. From their location in the vicinity of the RdRp termination codon, one possibility is that one or both of these hairpins are involved in protection against NMD, as NMD-protective sequences, if present, are located just downstream of a termination codon [11,67,68]. NMD is a eukaryotic mRNA surveillance pathway that detects and eliminates mRNAs containing premature termination codons, leading to lengthy 3′ UTRs. UPF1, the main NMD effector protein, binds to 3′ UTR sequences in translated RNAs with high G:C content [69] and/or 3′ UTRs of mRNAs that are longer than 200 nt in plants, which would include the gRNAs and sgRNAs of most plant RNA viruses [11,25]. Plant viruses known to resist NMD include betacarmovirus TCV, whose 200 nt 3′ UTR contains a 50 nt unstructured region near its 5′ end that is implicated in NMD resistance, and umbravirus PEMV2, whose ORF3 product (p26) protects viral RNA and a subset of NMD-targeted cellular transcripts, particularly those containing long, structured, GC-rich 3′ UTRs [11,50]. All ulaRNA gRNAs, which should terminate translation after ORF2, have 3′ UTRs greater than 200 nt (the 3′ UTR of CYVaV is 531 nt), suggesting that all should be susceptible to NMD in the absence of a protective *cis*-acting sequence or *trans*-acting protein. Since CYVaV accumulation in protoplasts was not enhanced in the presence of PEMV2 (and its encoded p26) (Figure 1D), it is possible that CYVaV may contain a *cis*-sequence that prevents NMD.

To determine if the 3′ UTR of CYVaV contains a *cis*-acting NMD-resistant sequence, *N. benthamiana* leaves were agroinfiltrated with GFP reporter constructs containing either the 3′ UTR of PEMV2 (construct GFP-3′UTRPEMV) or the 3′ UTR of CYVaV (construct GFP-3′UTRCYVaV) (Figure 10A). Constructs were infiltrated alone or together with a construct expressing U1D, a dominant-negative Upf1 inhibitor of NMD [25]. The 709 nt 3′ UTR of PEMV2 had no protective effect on NMD, and GFP levels increased 6-fold in the presence of U1D (i.e., when NMD was inhibited) (Figure 10B,C), as previously shown [11]. In contrast, GFP-3′UTRCYVaV transcripts were largely NMD-resistant, only increasing a modest 1.7-fold in in the presence of U1D. This strongly suggests that a cis-acting sequence in the 3′ UTR of CYVaV confers significant protection again NMD. If correct, then the strong positive effect of PEMV2 on CYVaV levels in *N. benthamiana* leaves may be for a reason other than NMD protection (Figure 1E). Since the 5′ region of 3′ UTRs in Class 2 ulaRNAs is highly structured without any sequence similarity, it will be interesting to determine precisely how this region confers NMD protection.

### 3.11. Features of Domain 2: Highly Conserved Structure of Unknown Function in CYVaV and OULV

Although D2 has less conserved sequence overall compared with D1, Structure 8 (beginning at position 1040) and surrounding sequences (total 150 nt) contains nearly 94% sequence identity between CYVaV and OULV (Appendix A). This region is not associated with the active site motif of the RdRp (GDD; position 1538 in CYVaV) and the reason for this strong sequence conservation is not known. Although all Class 2 ulaRNAs contain an identical or nearly identical short hairpin upstream of this region (Structure 7, Figure 3 and Figure 4), and an extended hairpin downstream (Structure 9, Figure 3 and Figure 4), Structure 8 and its strong sequence conservation is only found for CYVaV and OULV. Interestingly, EMaV and SULV share 84% sequence identity in this region and were predicted to contain a different structure that was also conserved (Figure 4). This region of structural conservation was not evident in Class 1 or Class 3 ulaRNAs.

### 3.12. Domain 3: Overall Structure

D3 contains the 3′ portion of the 3′ UTR and is the least conserved region among Class 2 ulaRNAs, with CYVaV sharing 44% sequence identity with OULV and 37% identity with EMaV. As described below, the SULV sequence is missing the 3′ terminal region of D3 and thus sequence identify was not determined. D3 begins with a series of short hairpins followed by a highly conserved structure containing short sequence stretches that are 100% conserved among Class 2 ulaRNAs (Structure 14) but not present in Class 1 and Class 3 ulaRNAs. At the 3′ end are elements that are similar to those found in most umbraviruses and carmoviruses. These features are detailed below.

### 3.13. Features of Domain 3: Structure 14

The most prominent feature of D3 is Structure 14 (Figure 11). The lower half of this structure was highly conserved among Class 2 ulaRNAs whereas the upper portion was variable. Using reporter constructs containing the 5′ proximal hairpin and the 3′ UTR, Structure 14 was identified as a 3′ cap-independent translation enhancer (3′ CITE) (J. Liu and A.E. Simon, in preparation). 3′ CITEs have been identified in an increasing number of viruses in the Tombusviridae, and function by interacting with various translation initiation factors to recruit 40S ribosomal subunits to the 3′ end of gRNA and sgRNA, which are subsequently transferred to the 5′ end mainly through RNA:RNA long-distance interactions [39,70]. Umbravirus PEMV2 contains three well studied 3′ CITEs (PTE, kl-TSS and TSS) [26,71,72,73], none of which resemble this element. Another 3′ CITE that is prevalent in umbraviruses (the BTE) also shares no sequence or structural similarity with Structure 14 [66]. The reported SULV sequence terminates within this 3′ CITE and thus the sequence appears to be truncated and not full-length as reported [20]. Structure 14 and all of its conserved sequences are absent in Class 1 and Class 3 ulaRNAs.

### 3.14. Features of Domain 3: 3′ Terminal Hairpins and Pseudoknot

Carmoviruses and umbraviruses contain conserved features at their 3′ ends including (from 3′ to 5′): a 3′ terminal hairpin known as Pr, a spacer region, the penultimate hairpin known as H5, and a pseudoknot (Ψ_1_) connecting sequence in an H5 internal symmetrical loop (carmoviruses) or terminal loop (umbraviruses) with the 3′ terminal four residues [40,73,74,75]. In addition, all of these viruses contain a hairpin just upstream of H5 known as H4b, whose terminal loop can frequently participate in an additional pseudoknot (Ψ_2_) with sequence just downstream of H5. TCV and cardamine chlorotic fleck virus (betacarmovirus) as well as five umbraviruses including PEMV2, also contain a hairpin adjacent to H4b known as H4a, which forms an H-type pseudoknot (Ψ_3_) with sequence upstream of H4a [66,73]. Altogether, H5, H4a, H4b, Ψ_2_ and Ψ_3_ form a ribosome-binding, T-shaped structure (TSS) that serves as a 3′ CITE [42,71,75] (Figure 12).

CYVaV and EMaV contain Pr, H5, Ψ_1_, H4b and H4a, whereas OULV is missing H4a (Figure 12). EMaV Ψ_1_ is unusual in not having the canonical GGGC/U:G/ACCC found in all carmoviruses and umbraviruses. None of the Class 2 ulaRNAs have the two pseudoknots necessary to form a TSS, and thus a TSS-type structure is unlikely to be present. The sequence complementary to the RSE sequence is located in the identical position between H5 and Pr in all Class 2 and Class 3 ulaRNAs. This is different from umbraviruses, where the sequence is always located in the Pr terminal loop. Although structures in comparable locations have not been identified for Class 1 ulaRNAs, Class 3 SbaVA contains Pr, H5, Ψ_1_ and H4b, similar to Class 2 ulaRNAs (Figure 12). In addition, the proposed long-distance interaction sequence with the SbaVA RSE is located in the identical location as in the Class 2 ulaRNAs. Whereas the 3′ ends of Class 2 and Class 3 ulaRNAs are structurally similar to umbraviruses, tlaRNAs have 3′ ends that strongly resemble tombusviruses (Appendix A), supporting the phylogenetic trees shown in Figure 2.

**Figure 12 viruses-13-00646-f012:**
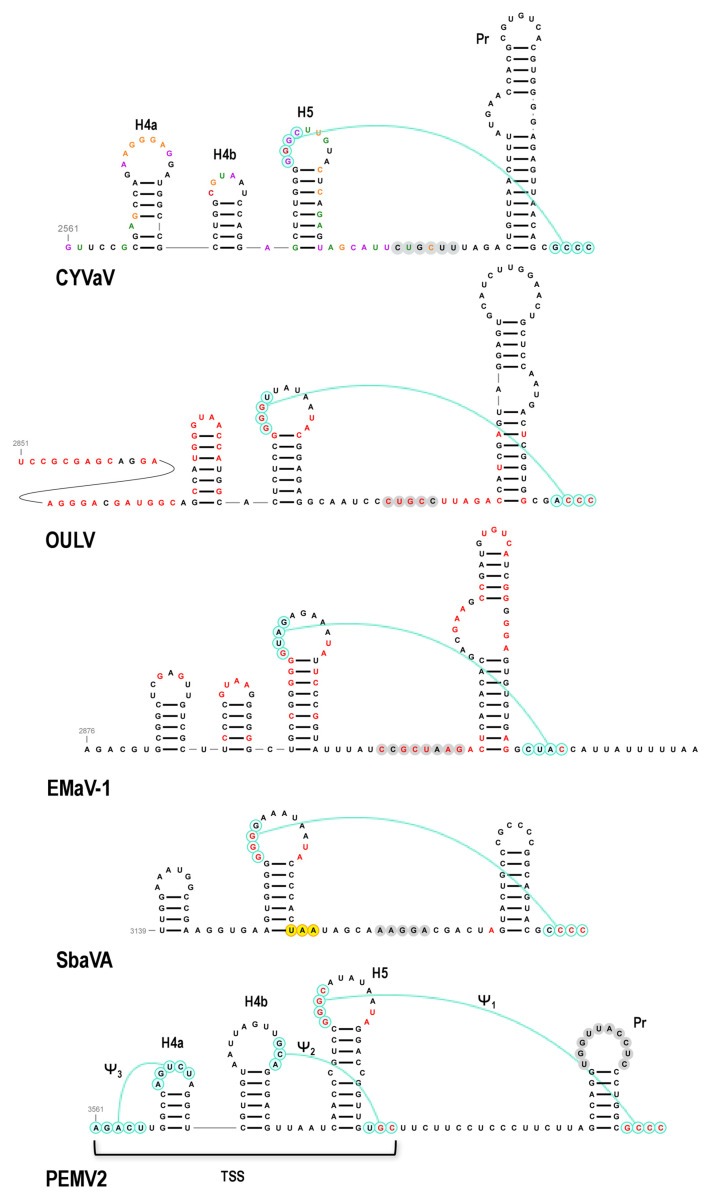
3′ terminal regions of Class 2 and Class 3 ulaRNAs and umbravirus PEMV2. Nomenclature for 3′ hairpins and pseudoknots were originally developed for TCV [75]. Three hairpins and 2 pseudoknots comprise a ribosome-binding, T-shaped structure (TSS) that serves as one of three 3′ CITEs in PEMV2 and is structurally conserved in many umbraviruses and two betacarmoviruses. CYVaV and EMaV contain the corresponding hairpins of the TSS but not the pseudoknots. Since all umbraviruses and carmoviruses contain Ψ_1_ at their 3′ termini, it is unclear if the additional downstream sequence in EMaV is correct. The termination codon for ORF5-1 in SbaVA is in yellow.

### 3.15. Evolutionary Relationships among ulaRNAs Based on Structural Domains

Unlike other viruses where selection pressure for different proteins could be determined [76,77], the different lengths of ulaRNA RdRp sequences and the smaller number of sequences available makes it challenging to conduct proper tests to assay for selection pressure on the RdRp. Based on the presence of the RSE within the RdRp ORF, the CCS at the 5′ ends of the gRNA and presumptive sgRNA, and the conserved 3′ terminal structural elements found in umbraviruses and both Class 2 and Class 3 ulaRNAs, it is reasonable to speculate that ulaRNAs originated from the same ancestral virus as umbraviruses, which agrees with the phylogenetic trees based on RdRp ORF nucleotide and aa sequence (Figure 2A,B). However, a lower number of nucleotide substitutions per site (Figure 2B) than protein substitutions per site (Figure 2A) indicates that the nucleotide sequence is more conserved than the aa sequence, suggesting an evolutionary need to conserve specific RNA structures. These substitution variations suggest that cladograms based on conserved nucleotide structural domains might yield additional information to help in understanding the evolutionary relationships among ulaRNAs and between ulaRNAs and umbraviruses.

To investigate the evolutionary relationships based on conserved Class 2 RNA structures, cladograms were generated for the nucleotide sequences of groups of structures in relation to related regions from umbraviruses and Class 1 and Class 3 ulaRNAs. Group 1 consisted of Structures 1–6 (Figure 13A), Group 2 contained Structures 7–9 (Figure 13B) and Group 3 contained Structures 12–14 (Figure 13C). In all cladograms irrespective of different groups, CYVaV and OULV made a single clade with moderate to strong bootstrap support of at least 62%, suggesting that they are more genetically related among the ulaRNAs, in line with the previous phylogenetic trees (i.e., Figure 2B, encompassing RNA structures 1–11). Class 1 ulaRNAs made a monophyletic clade with strong support (100%) in trees based on Group 1 and Group 2 while failing to make a significant clade in the tree based on Group 3. This suggested that there was less genetic relatedness in its 3′ UTR sequence, which agrees with our observation that when compared with Class 2 ulaRNAs, there were no discernable conserved sequences or structures in this region in Class 1 ulaRNAs. Interestingly, the grouping of Class 3 SbaVA with Class 1 ulaRNAs was only strongly supported in the tree based on Group 2 structures (Figure 13B). Its weak relationship with Class 1 and Class 2 ulaRNAs in the trees based on Group 3 and Group 1 structures, respectively, further support our placement of SbaVA in a separate class (Class 3).

Considering all the trees together (Figure 2 and Figure 13), we propose a model for the evolution of ulaRNAs and umbraviruses (Figure 14). We suggest that the progenitor was an ulaRNA-like replicon that required a helper virus for transmission. Two different recombination events, involving either viral or host mRNAs encoding RNA movement proteins [78], gave rise to umbraviruses and Class 2 ulaRNAs. We suggest that CYVaV diverged from other Class 2 ulaRNAs by loss of two segments, along with alterations in the RdRp ORF sequence that previously overlapped with that of ORF5 (e.g., loss of the ORF5 initiation codon and the CCS, and accumulation of stop codons in the previous ORF), and other changes that were necessary to stabilize the remaining CYVaV sequence. We hypothesize that host selection may have mediated the deletion of the two fragments (and loss of the ORF5 encoded product) based on the previous demonstration for tobacco mild green mosaic virus (tobamovirus), which often has a duplicated sequence in its 3′ UTR that contributes to fitness in one host but not another [79]. If true for CYVaV, this would imply that in citrus, CYVaV does not need the ORF5 presumptive movement protein function (Appendix A). This product, however, may still be beneficial in opuntia, maize and sugarcane.

**Figure 13 viruses-13-00646-f013:**
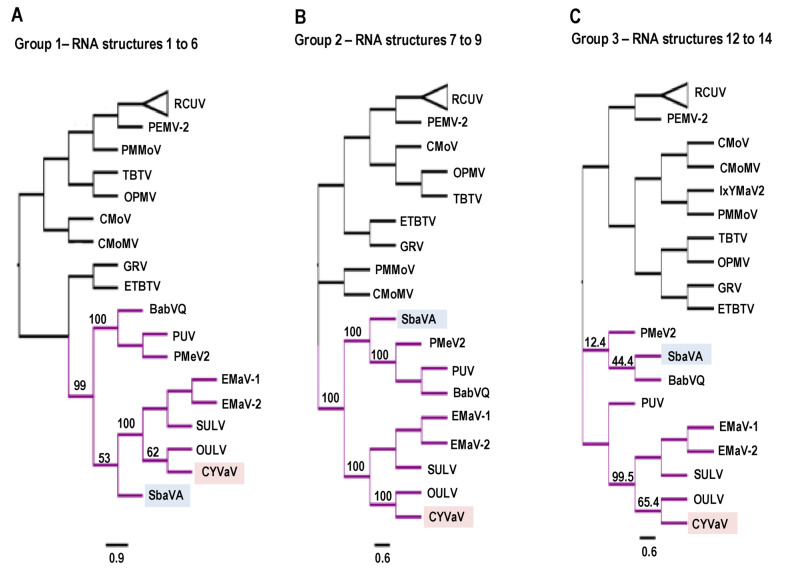
Cladograms of maximum likelihood trees based on the nucleotide sequence of umbravirus and ulaRNA regions. (**A**) Group 1: RNA structures 1–6. (**B**) Group 2: RNA structures 7–9. (**C**) Group 3: RNA structures 12–14 (see Figure 4). The above branch numbers indicate bootstrap support in percentage out of 1000 replicates. All the trees were mid-point rooted. See legend to Figure 2 for abbreviations.

The current model also accounts for the grouping together of Class 1 and Class 3 ulaRNAs based on RdRp sequence by suggesting that they originated from a common ancestor replicon. Based on weak homology between corresponding regions of Class 2 ORF5 and Class 3 ORF5-1, along with their different translation initiation and termination sites, we proposed that ORF5-1 in Class 3 SbaVA was acquired by a separate recombination event, which led to the divergence of Class 1 and Class 3 ulaRNAs. Acquisition of a more extensive ORF by a separate event would also account for the lack of Structures 12–14 in SbaVA. This model lays down a road map for future investigations into these novel virus-like RNAs.

## Figures and Tables

**Figure 3 viruses-13-00646-f003:**
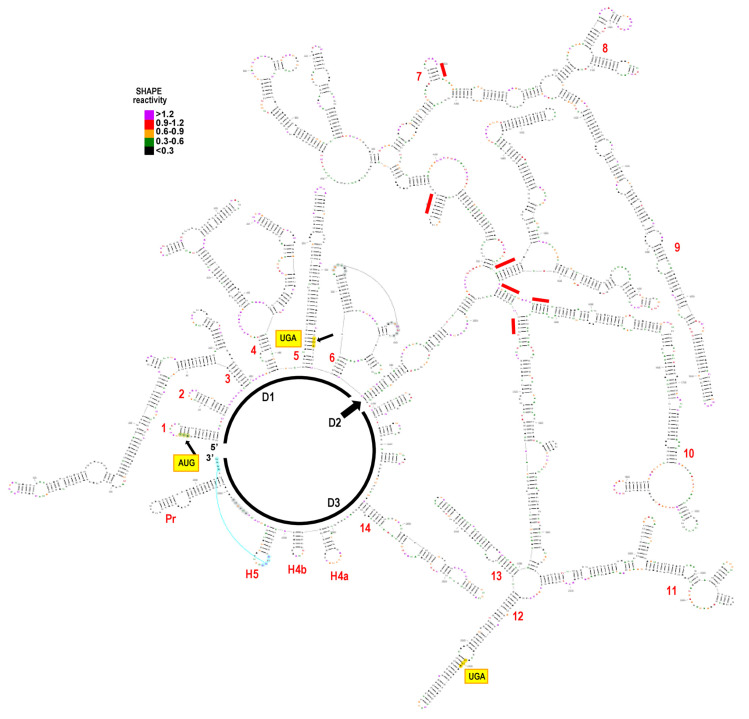
Proposed RNA structure for full-length CYVaV. CYVaV transcripts were synthesized using T7 RNA polymerase, denatured, snap cooled and then treated with NMIA or DMSO as described in the Materials and Methods. Ten primers labeled with 6FAM were used for reverse transcription of the SHAPE modified samples and PET was used for sequencing ladder samples. Data that was obtained from 2 to 3 repeats of the primer sets were analyzed using QuSHAPE software [28]. Colors denote SHAPE reactivity, with purple the most reactive, followed by red, orange, green and black being the least reactive. The structure was divided into three domains (D1, D2 and D3) for ease of presentation. Structures referred to in the text are numbered in red. Red lines denote key base-paired helices that were highly conserved in both sequence and structure among the Class 2 ulaRNAs, and that were important in conceptualizing the final structure. The location of the initiation codon for p21, p21 termination codon (UGA) and p81 termination codon UGA are shown. Two putative tertiary interactions (see text) are denoted by curved lines. Note that subsequent figures show enlargement of key regions of the structure and SHAPE data.

**Figure 4 viruses-13-00646-f004:**
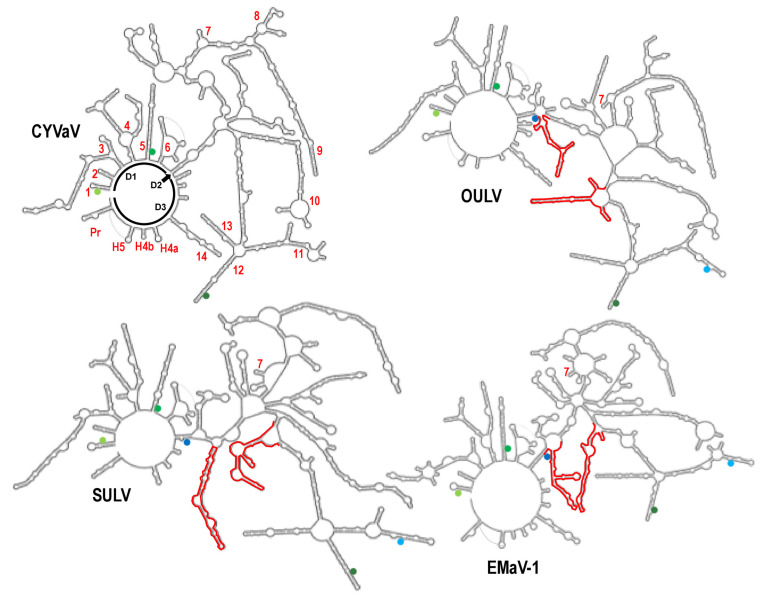
Comparison of the CYVaV RNA structure with proposed structures for other Class 2 ulaRNAs. See Figure 3 legend for explanation of numbers and domains shown for CYVaV. Additional designations of CYVaV structures (Pr, H5, H4a and H4b) can be found in Figure 12. Structure 7 (small hairpin) is highly conserved and is denoted for each genome structure. Inserted segments not found in CYVaV are shown in red. Light green, medium green and dark green circles denote ORF1 initiation site, ORF1 termination site and ORF2 termination site, respectively. Light blue and medium blue circles denote start site and termination site for ORF5, respectively.

**Figure 5 viruses-13-00646-f005:**
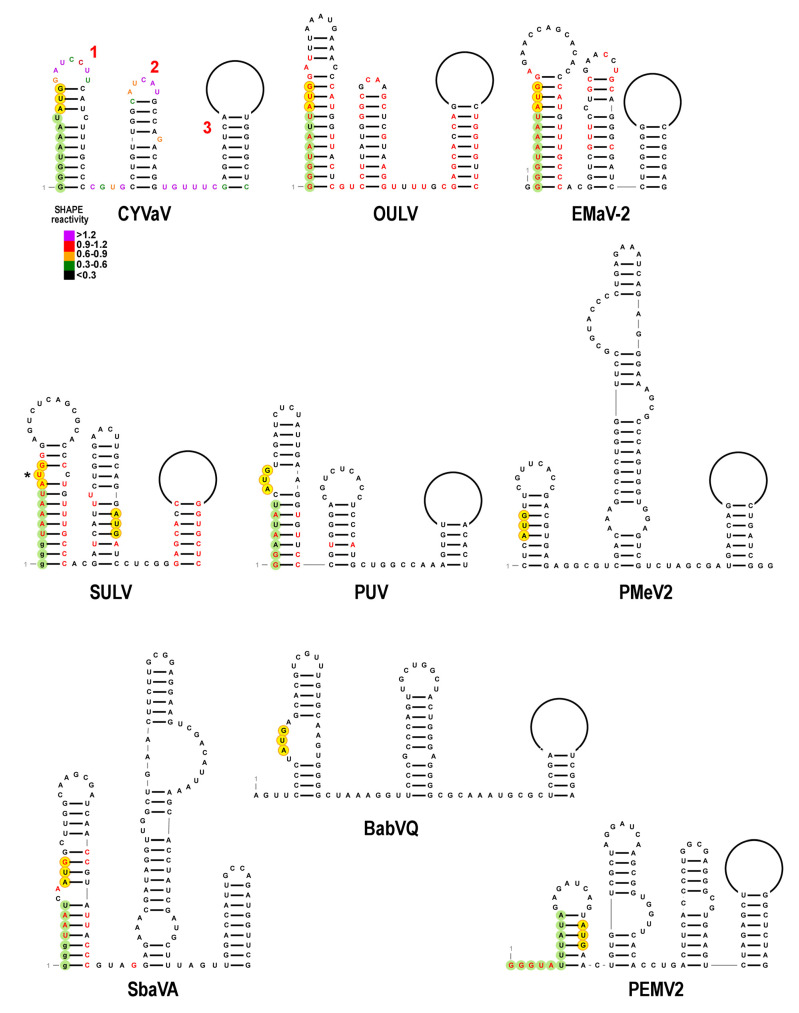
Structures at the 5′ end of the ulaRNAs and umbravirus PEMV2. Three hairpin CYVaV structure that was predicted using SHAPE structure probing is shown with proposed structures for the other ulaRNAs. SHAPE data is presented on the CYVaV structure whereas residues in red in the other structures denote sequence conservation with CYVaV. The carmovirus consensus sequence (CCS) is shaded green and initiation codon is shaded yellow. CYVaV stems are numbered according to Figure 3. Asterisk denotes first AUG codon in SULV that is out of frame with most of ORF1. The second AUG is proposed to be the initiation codon for SULV p20. Note that for Class 1 ulaRNAs, PUV, but not PMeV2 or BabVQ, contains a CCS at its 5′ end. Guanylates that are absent in reported SbaVA and SULV sequences (but may be present) are in -lower case.

**Figure 9 viruses-13-00646-f009:**
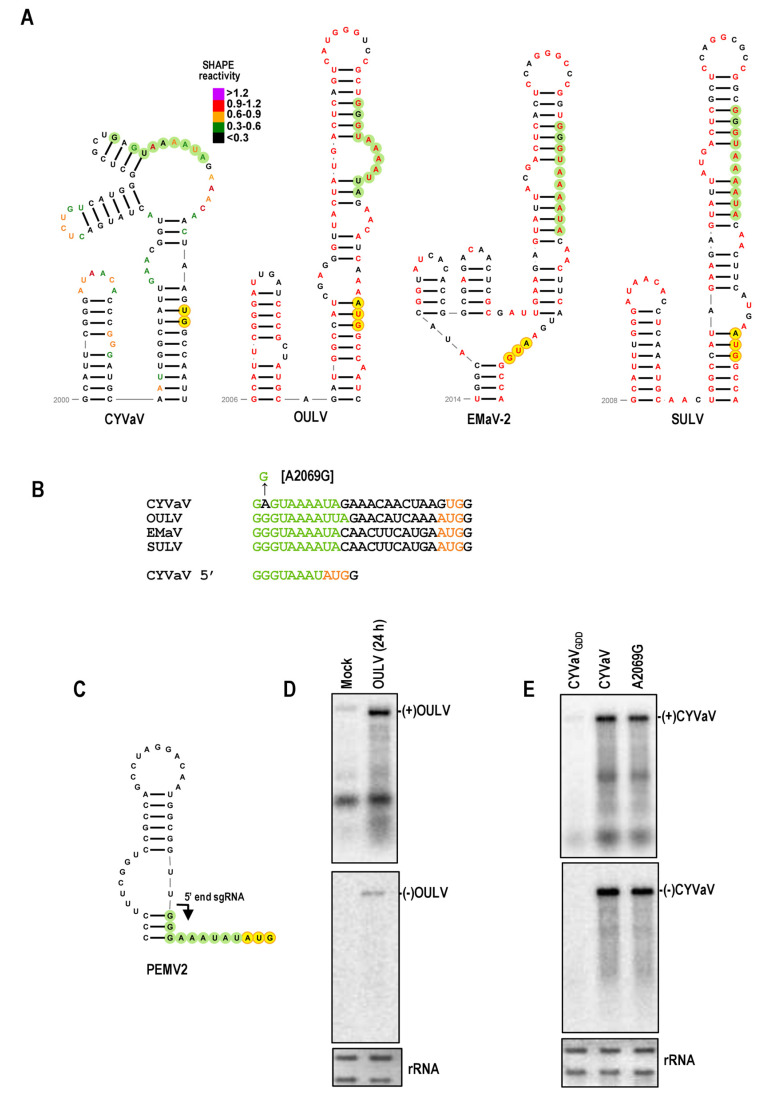
Comparison of the region encompassing the ORF5 start site. (**A**) SHAPE-derived Structure 11 in CYVaV and proposed structures for the other Class 2 ulaRNAs. SHAPE data is shown for the CYVaV structure. Red residues in the other proposed structures are conserved with CYVaV. CCS sequence is shaded green. Note that the CYVaV CCS is missing a conserved G residue. Translation initiation site for ORF5 is shaded yellow in OULV, EMaV and SULV. In CYVaV, the sequence is a “GUG” and the UG are shaded yellow. (**B**) Sequence of the Class 2 CCS (shown in A) compared with the 5′ end of CYVaV (5′). Base alteration in CYVaV mutant A2069G that generated a canonical CCS is shown. (**C**) Conserved structure in the vicinity of the mapped sgRNA from PEMV2 [66]. (**D**) Northern blot analysis of wild-type OULV accumulating in *Arabidopsis* protoplasts inoculated with full-length transcripts. OULV RNA levels were examined at 24 h post-inoculation by probing for (+)OULV (top) and (−)OULV (bottom). Ethidium bromide-stained 28S rRNA was used as a loading control. (**E**) Northern blot analysis of CYVaV wild-type (CYVaV) and mutant transcripts. CYVaV_GDD_, CYVaV with a mutation in the RdRp active site. Alternation in A2069G is shown in (**B**). No sgRNA product was discernable for OULV or wild-type or mutant CYVaV.

**Figure 10 viruses-13-00646-f010:**
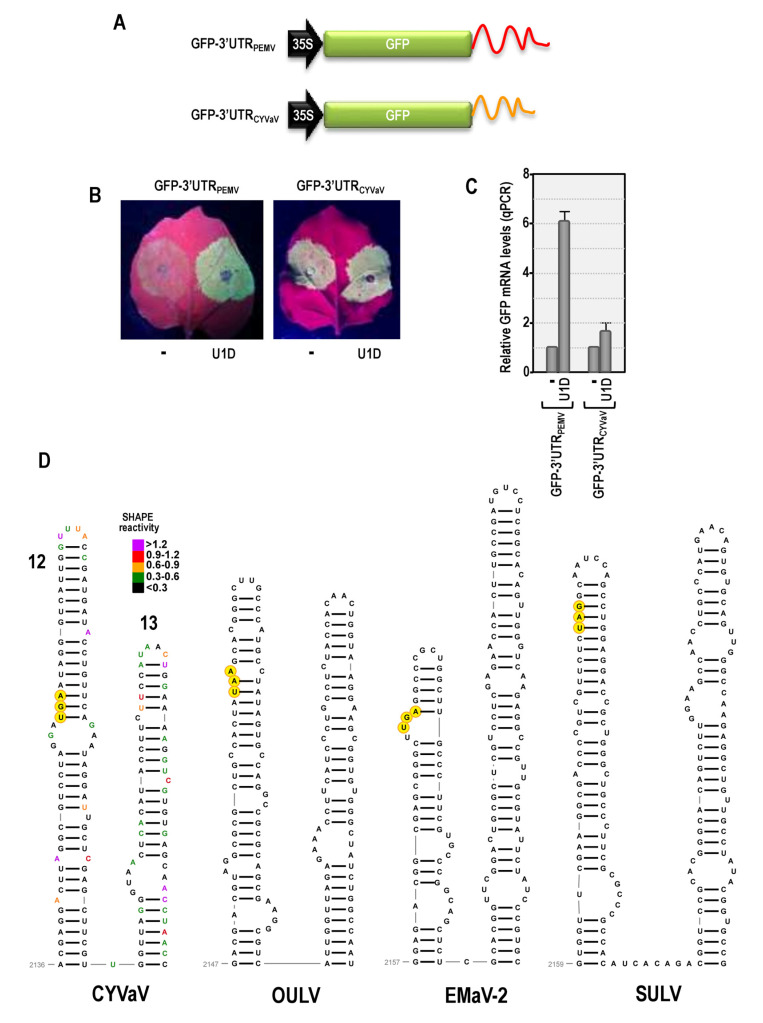
CYVaV 3′ UTR protects susceptible transcript from NMD. (**A**) Constructs used in this analysis. GFP-3′UTR_CYVaV_ is an expression cassette containing the 35S promoter from CaMV followed by the GFP ORF, full-length CYVaV 3′ UTR, and a ribozyme designed to generate transcripts that terminate at the end of the 3′ UTR. (**B**) *N. benthamiana* leaves were infiltrated with *A. tumefaciens* containing a Ti-plasmid with either the PEMV2 or CYVaV cassette (shown on left side of leaf) either alone or coinfiltrated with a Ti-plasmid expressing U1D, a dominant negative inhibitor of NMD (shown on right on the leaves). GFP levels were visually inspected 5-days post-infiltration. (**C**) Infiltration “spots” were excised and assayed for GFP mRNA levels by qPCR. Data represents three independent experiments and standard deviation is shown. (**D**) Conserved structures 12 and 13 at the 3′ end of the RdRp ORF (ORF2). SHAPE data is shown for the CYVaV structures. No significant sequence conservation is found in the other ulaRNA structures. The stop codon for ORF2 is shaded yellow.

**Figure 11 viruses-13-00646-f011:**
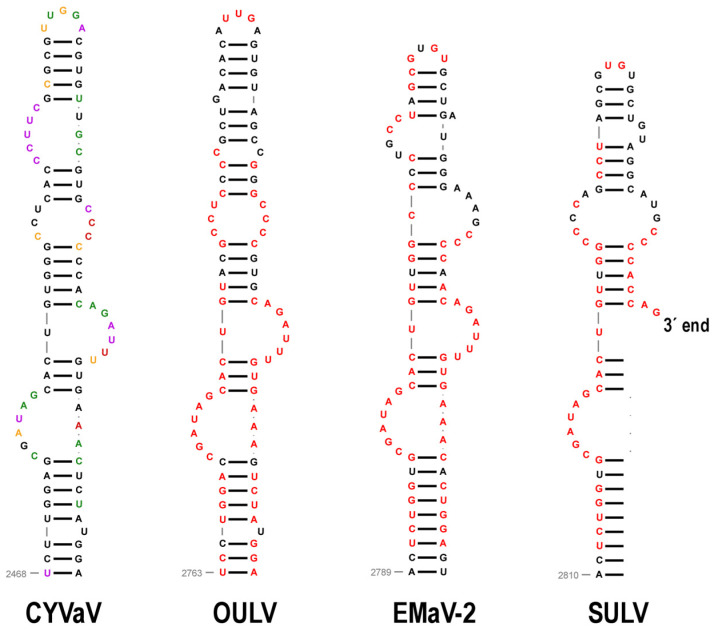
Structure 14 in Class 2 ulaRNAs. This structure has been identified as a 3′ CITE (J. Liu and A.E. Simon, unpublished). SHAPE data is shown on the CYVaV structure. Red bases in the other ulaRNA Structure 14 are identical to those of CYVaV. The reported SULV sequence terminates within this structure and thus the sequence is unlikely to be full-length as reported [20]. This element is not conserved in the Class 1 and Class 3 ulaRNAs, which must contain different 3′ CITEs.

**Figure 14 viruses-13-00646-f014:**
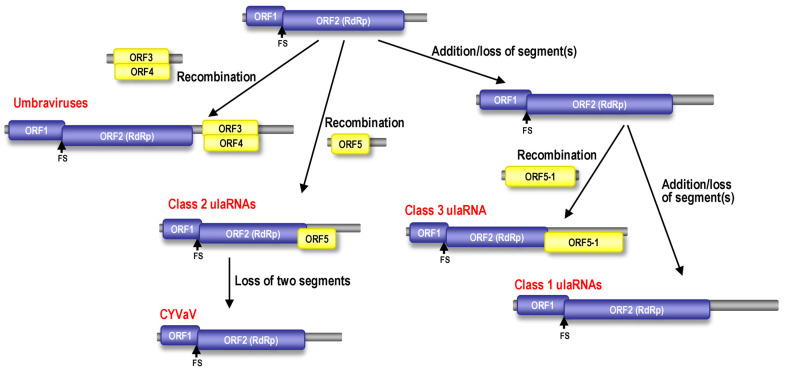
Model for the evolution of ulaRNAs and umbraviruses. See text for explanation. FS, -1PRF site.

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
