# Peer review of "Structural Analysis and Whole Genome Mapping of a New Type of Plant Virus Subviral RNA: Umbravirus-Like Associated RNAs"

_viruses, 2021, doi:10.3390/v13040646_

Round 1

Reviewer 1 Report

The MS titled “Structural Analysis and Whole Genome Mapping of a New Type of Plant Virus Subviral RNA: Umbravirus-Like Associated RNAs” reports on the biological and structural characterization of umbravirus-like associated  RNAs (ulaRNAs).

First of all, I thank the authors for forcing me to read about ulaRNAs and certainly refreshing to review a molecular virology heavy manuscript. The MS is overall well written and I do not see any issues with the experimental work and the interpretation of the results. The authors present a good and solid introduction. I recommend that any reader of this MS to please grab a cup of tea or a mug of coffee, and be prepared to spend some time reading the MS properly.

Below are some minor comments to help improve the MS

  • Line 44 and through out the MS: please note that species names cannot be abbreviates. Thus I suggest here your refer to the virus itself so it would be “turnip crinkle virus (TCV)”. For virus names the first letter is not capital.
  • Line 51: same as above should be “pea enatation mosaic virus 2 (PEMV2)”
  • Line 83: “papaya ringspot virus”
  • Line 95-97: change to “Similar ulaRNAs sequences from sugarcane (sugarcane umbra-like virus; SULV) (19) and citrus (citrus yellow vein associated virus; CYVaV) (Kwon et al., submitted) have been reported, and additional sequences from Ethiopian maize (Ethiopian maize associated virus; EMaV) and strawberry (strawberry associated virus A; SbaVA) have been deposited in GenBank (Fig. 1).”
  • Line 139 and throughout the MS: Please note that the that Rhizobium radiobacter is the updated name for Agrobacterium radiobacter
  • Line 230: change to “viruses in the family Tombusviridae
  • Line 247: change to “ thaliana
  • Line 251: change to “Northern blot analysis”
  • Line 261: “tombusvirus tomato bushy stunt virus (TBSV)”
  • Line 290: “barley yellow dwarf virus (BYDV)”
  • Line 292: change suggested to revealed. The software does not suggest but the certainly the data derived from it can be suggestive of…
  • Line 293: “pelargonium leaf curl virus (PLCV)”
  • Line 321-328: here you are referring to the viruses (GenBank accession #s are for individual viruses) thus not species. Please edits to remove italics and the first letter is not capital expect in the case of Ethiopian.
  • Line 387: please provide an accession # for SULV as you mention GenBank.
  • Line 412: edit to “throughout the sequences of the members of the Tombusviridae family,…”
  • Line 646: Figure 13 – these appear to be cladograms and not phylograms. Note the step like branches with equal length. Please check an edit.

Author Response

First of all, I thank the authors for forcing me to read about ulaRNAs and certainly refreshing to review a molecular virology heavy manuscript. The MS is overall well written and I do not see any issues with the experimental work and the interpretation of the results. The authors present a good and solid introduction. I recommend that any reader of this MS to please grab a cup of tea or a mug of coffee, and be prepared to spend some time reading the MS properly.

 A can of diet coke would also be an appropriate beverage.  This was a very unusual manuscript to write and we are delighted that the reviewer appreciated it.

  • Line 44 and through out the MS: please note that species names cannot be abbreviates. Thus I suggest here your refer to the virus itself so it would be “turnip crinkle virus (TCV)”. For virus names the first letter is not capital.

Corrected throughout

  • Line 51: same as above should be “pea enatation mosaic virus 2 (PEMV2)”

Corrected

  • Line 83: “papaya ringspot virus”

Corrected

  • Line 95-97: change to “Similar ulaRNAs sequences from sugarcane (sugarcane umbra-like virus; SULV) (19) and citrus (citrus yellow vein associated virus; CYVaV) (Kwon et al., submitted) have been reported, and additional sequences from Ethiopian maize (Ethiopian maize associated virus; EMaV) and strawberry (strawberry associated virus A; SbaVA) have been deposited in GenBank (Fig. 1).”

Corrected

  • Line 139 and throughout the MS: Please note that the that Rhizobium radiobacter is the updated name for Agrobacterium radiobacter

That may be the updated name 20 years ago, but I (and virtually everyone else in plant molecular biology/virology) refuse to use it as it will cause immense confusion to readers. Agroinfiltration has been used as a term for thousands of papers over 30 years too.

We did all this sentence to the Materials and Methods: Agrobacterium tumefaciens was renamed Rhizobium radiobacter but the original name will be used here for clarity.

  • Line 230: change to “viruses in the family Tombusviridae

Corrected, and made same changes to the Luteoviridae

  • Line 247: change to “ thaliana

Not sure what the reviewer wants.  Current line is:  Arabidopsis thaliana protoplasts were inoculated with in vitro synthesized CYVaV transcripts

  • Line 251: change to “Northern blot analysis”

Corrected

  • Line 261: “tombusvirus tomato bushy stunt virus (TBSV)”

Corrected

  • Line 290: “barley yellow dwarf virus (BYDV)”

Corrected

  • Line 292: change suggested to revealed. The software does not suggest but the certainly the data derived from it can be suggestive of…

Thanks!  Corrected

  • Line 293: “pelargonium leaf curl virus (PLCV)”

Corrected

  • Line 321-328: here you are referring to the viruses (GenBank accession #s are for individual viruses) thus not species. Please edits to remove italics and the first letter is not capital expect in the case of Ethiopian.

Corrected

  • Line 387: please provide an accession # for SULV as you mention GenBank.

It was provided on line 329.  StaVA was left out and the accession number is now provided.

  • Line 412: edit to “throughout the sequences of the members of the Tombusviridae family,…”

Changed

  • Line 646: Figure 13 – these appear to be cladograms and not phylograms. Note the step like branches with equal length. Please check an edit.

Changed to cladogram here and in the text

Reviewer 2 Report

The work of Liu and co-workers presents a comprehensive analysis on the  structure  and the Genome Mapping of  Umbravirus-Like Associated RNAs. The presentation although (inevitably) rather complex in some sections,  is well  written, the experiments well designed and executed. The manuscript is recommended for publication in Viruses. 

Author Response

We agree with the reviewer.